# Plasmonic Nanofactors as Switchable Devices to Promote or Inhibit Neuronal Activity and Function

**DOI:** 10.3390/nano9071029

**Published:** 2019-07-18

**Authors:** Karrer M. Alghazali, Rabab N. Hamzah, Zeid A. Nima, Richard Steiner, Madhu Dhar, David E. Anderson, Abdallah Hayar, Robert J. Griffin, Alexandru S. Biris

**Affiliations:** 1Center for Integrative Nanotechnology Sciences, University of Arkansas at Little Rock, Little Rock, AR 72204, USA; 2Department of Radiation Oncology, University of Arkansas for Medical Sciences, Little Rock, AR 72205, USA; 3Tissue Regeneration Laboratory, Department of Large Animal Sciences, College of Veterinary Medicine, University of Tennessee, Knoxville, TN 37996, USA; 4Department of Neurobiology and Developmental Sciences, University of Arkansas for Medical Sciences, Little Rock, AR 72205, USA

**Keywords:** gold nanosystems, neuronal activity, synthetic extracellular matrices, plasmonic

## Abstract

Gold nanosystems have been investigated extensively for a variety of applications, from specific cancer cell targeting to tissue regeneration. Specifically, a recent and exciting focus has been the gold nanosystems’ interface with neuronal biology. Researchers are investigating the ability to use these systems neuronal applications ranging from the enhancement of stem cell differentiation and therapy to stimulation or inhibition of neuronal activity. Most of these new areas of research are based on the integration of the plasmonic properties of such nanosystems into complex synthetic extracellular matrices (ECM) that can interact and affect positively the activity of neuronal cells. Therefore, the ability to integrate the plasmonic properties of these nanoparticles into multidimensional and morphological structures to support cellular proliferation and activity is potentially of great interest, particularly to address medical conditions that are currently not fully treatable. This review discusses some of the promising developments and unique capabilities offered by the integration of plasmonic nanosystems into morphologically complex ECM devices, designed to control and study the activity of neuronal cells.

## 1. Introduction

Tissue damage and organ failure can occur in a number of reasons, including trauma, infection, disease, or various degenerative conditions (Alzheimer’s, macular degeneration, and osteoporosis). Trauma and infection occur in individuals as a result of hazards experienced in civilian life, work-place injuries, intense training, military engagements, and terrorist activities. Depending on the degree of damage and healing properties, the tissue’s integrity and function may be able to be restored through regenerative processes. Based on the type of tissue, different types of cells can be introduced to enhance the recovery process. Most recently, research has focused on adult mesenchymal stem cells as a potential biological therapy for tissue repair applications. Overall, recovery is a function of the ability of stem cells to survive, proliferate, and differentiate into a lineage capable of repairing the damaged tissue. Stem cells also may serve to recruit endogenous cells to the injury site and stimulate these cells to enhance tissue regeneration. The key element for these various types of cells to survive and enhance healing of inured tissues is the microenvironment and ability to develop structures to support proliferation. The microenvironment (scaffolds or extracellular matrix (ECM)) is a key factor in regulating the self-renewal and differentiation of stem cells [1,2]. Generally, neural tissue ECM consists of different proteins such as, laminin (LN), perlecan (PN), collagen, nidogen, and fibronectin (Fn) [3]. Some of these proteins, primarily the fibronectin and collagen combine to form a unique substrate structure, the basal lamina (BL). This unique substrate promotes cell attachment, proliferation, and migration. In addition to providing physical support, the ECM has distinct components that can bind to cell integrins to stimulate signaling pathways and modify the cytoskeletal organization of the cell. This ECM-cell interaction can guide cell migration by forming focal adhesions and regulating cell morphology [4,5]. Biochemical and biophysical boundaries are the main factors governed by the ECM. Optimizing these boundaries during tissue recovery would help promote a healthy microenvironment that is ultimately required for new tissue formation and regeneration. Different approaches have been used to mimic and enhance tissue regeneration by creating supportive nano- and micro-sized artificial ECM in the form of scaffolds or substrates. The choice of approach depends on the type of tissue and the degree of tissue injury. Artificial ECMs have been developed using fiber gels, nanoparticles polymers, ceramics, and other materials [6,7,8]. Artificial ECMs can have unique features, such as personalized functional groups, degrees of hydrophilicity, electrical conductivity, and optical activity [9,10,11,12,13]. The main considerations for the development of novel ECM materials rely on their physico-chemical and mechanical properties [such as surface chemistry, surface energy, roughness, morphology, mechanical rigidity, and robustness) that ultimately control their biocompatibility and long-term tissue integration. Also, the ability of the ECM to eventually degrade over a chosen period of time and result in limited inflammatory responses and toxic bi-products is preferred.

The most common artificial ECM are polymer-based (natural or synthetic), with the intent to use such material to accelerate the rate of nerve recovery, during or after injury. For example, collagen-based artificial ECMs are used and studied because they are biocompatible and present in various connective tissues [14]. The collagen-based artificial ECMs have been integrated into nerve conduits or wraps, with the main goal of accelerating the rate of damaged tissue recovery [14]. Polymer-based artificial ECMs can enhance recovery of nerve injury but it is not clear whether these materials can lead to improvement in the characteristics of regenerated tissue.

Recently, a new method to treat tissue injuries has been proposed [15]. It utilizes a remotely activated ECM that can be applied specifically to enhance the generation of conductive tissue. The basic components of this artificial ECM are plasmonic nanoparticles that are activated by exposure to laser-electromagnetic waves. In theory, laser-electromagnetic waves trigger events that modify cell function. The surface chemistry of these plasmonic nanofactors can be tuned [16], which makes them ideal for construction of an artificial ECM.

Exploiting the interaction between cells and plasmonic nanofactors (PNFs) is one of the promising emerging methods for altering cell membrane potential and, subsequently, neuronal excitability. PNFs can function as a non-invasive, tunable “switch” to turn on or off the activity of neurons. The basic principle of this technique is to create a plasmonically active surface close to the neural cell; the cell properties can be altered by this surface, including modulation of the cell’s electrical activity. The development of PNFs that can affect neural activity should be based on an understanding of the events and factors that control the electrophysiological properties of neurons. Potentially, this technology could be used to control neural activation or to treat neural injury. Several factors must be considered to improve this technology, such as the ability to promote or inhibit neural activity, the nature of the surface-cell interactions, biocompatibility of the surfaces, the ability to use these surfaces to affect neural activity in a reproducible manner (i.e., without neuronal adaptation in response to the stimulus), and how to deliver or embed such surfaces in vivo [9,15,17,18,19,20,21,22,23].

A commonly studied type of PNFs is represented by the gold nanorods (GNRs), with their surface plasmon resonance taking place also within the near infrared (NIR) region. This unique feature makes GNRs a valuable tool for biomedical applications, as biological tissue is maximally transparent in the NIR wavelength. As such, GNRs may permit deep tissue stimulation [22,24]. Surface plasmon resonance is generated when an electromagnetic wave strikes a metal surface at a frequency that matches that of the oscillations of free electrons. This phenomenon has been shown to excite the electrons within a region that is considerably close to the surface. These electrons reflect the gaining energy as a resonant oscillation over the surface (Figure 1) [25,26].

This review discusses the potential use of PNFs to modulate neural activity, either as an enhancer or as an inhibitor. The first type of investigation explores the conditions under which PNF interaction with the cell membrane enhances neural activity, differentiation of stem cells into neural lineage cells, and neural differentiated cell growth. A second research direction explores whether cell-PNF interactions can inhibit neural network electrical activities, such as by causing membrane potential hyperpolarization and decreasing activation frequency.

## 2. Plasmonic Nanofactors as Neural Simulators

Although a relatively new area of research, several studies have investigated the use of plasmonic surfaces as an extracellular support for neural cells and examined the effects of activated surface plasmons on neural activity. The prevailing interpretation for the effect that PNFs have on neural cells is based on the energy emitted by resonant oscillation. During resonant oscillation, excitation of the plasmonic surface may cause transient heating. This form of energy stimulates the cell, causing a change in the membrane capacitance and membrane depolarization which triggers the cell to fire action potentials (Figure 2) [19,27,28].

Utilizing PNFs is considered an innovative approach for modulating cellular activity based on cell-plasmon or cell-surface interaction [9,22,23,29,30,31,32,33,34,35,36]. Past in vitro experiments have exposed cells to a plasmonic nanomaterial dispersed in the cell culture medium or to a plasmonic nanomaterial deposited as 2D structures. In contrast, in vivo experiments have injected the PNFs into the injured area [15]. Knowledge gained by these studies aimed to investigate the effects of the presence of PNFs on the cell microenvironment. Additionally, data were provided on the establishment of plasmonic nanofactor-laser simulation parameters that could potentially be adapted for future investigations [9,15,22,23,29,30,31,32,33,34,37].

Paviolo et al. [30] demonstrated that exposure of NG108-15 neuronal cells to both pure AuNRs and AuNRs coated with different structures, dispersed in the cell culture medium led to the activation of their surface plasmon after exposure to a 1.2–7.5 W/cm^2^ laser at 780 nm. The study concluded that the surface chemistry of AuNRs promotes neurite formation after four days of culture. However, cells incubated with and without AuNRs had an initial noticeable decrease in cell viability under 780-nm laser diode exposure at irradiances of at least 250 mW/cm^2^. This cell viability trend was only seen during the first 24 h, and the researchers attributed it to the effect of laser irradiation potentially causing cell damage. The activation of AuNRs’ surface plasmon positively enhanced neuronal cell differentiation when laser power was increased. A 36% increase in average neurite length was seen compared to non-irradiated cells. This behavior was attributed to the increase in reactive oxygen species resulting from transient heat generated by localized surface plasmon resonance. This resonance may activate transcription factors that enhance cell metabolic pathways.

Eom at al. [33] developed a local AuNR delivery system, based on surface-modified AuNRs, to investigate the effects of the activation of the PNF surface on intracellular Ca^2+^ transients in astrocytes. To ensure the precise delivery of the AuNRs to the astrocyte plasma membrane, the AuNR surface was decorated with streptavidin. This tagging system relies on the tendency of biotinylated anti-Thy-1 antibodies to conjugate with thymocyte antigen 1 (Thy-1) on the surface of astrocytes. Thy-1 is an astrocyte–specific protein commonly expressed this type of cells and hence, can be used as a target/marker protein. The IR stimulation system implemented throughout this study involves a laser controller, an optical fiber (diameter: 600 μm; numerical aperture: 0.14) coupled with a laser diode (980 nm), and a stimulation probe, as shown in Figure 3. To ensure precise laser illumination, the incident laser beam’s position was determined by a red guiding beam through the optical microscope.

In Eom et al.’s study, the stimulation protocol utilized a single, short NIR pulse (duration: 950 µs) and a power of 13.12 mJ/mm^2^ to produce intracellular Ca^2+^ waves in astrocytes, and the intracellular Ca^2+^ movement was monitored using fluorescence microscopy and Rhod-2, a Ca^2+^ indicator. Monitoring Ca^2+^ ion concentration in cultured astrocytes for NIR stimulation in the presence of membrane-bound AuNRs and the control, which lacked targeting molecules bound to AuNRs, revealed Ca^2+^ waves associated with the membrane-bound AuNRs after optical stimulation compared to controls, as shown in Figure 4. The study concluded that the local heat shock generated as a result of surface plasmon activation caused asymmetrical movement of ions across the membrane lipid bilayer, which induced a net charge displacement across the membrane and activation of voltage-gated Ca^2+^ channels.

Changes in cellular electrophysiology after exposure to PNFs were also investigated by Yong et al. [23]. The researchers monitored the electrical activity of primary auditory neurons in response to PNF activation. The experimental setup, shown in Figure 5, involved in vitro exposure of primary auditory neurons to silica-coated AuNRs and silica-coated gold nanospheres, followed by testing the effect of PNF stimulation on the electrical properties of neurons using whole-cell patch clamp electrophysiology. The PNF was activated by a 780-nm laser diode with constant power of 90 mW. The laser was linked to a 125-μm (8.2-μm core) optical fiber for band expansion; for accurate light stimulation, the laser was aligned with the target cell body using a micromanipulator. Before PNF stimulation, a depolarizing current was injected to examine the ability of primary auditory neurons to fire action potentials in the presence or absence of PNF.

The patch-clamp recording showed unique waveform generation in the presence of AuNRs compared to silica-coated gold nanospheres and the control (cell alone). A noticeable difference in membrane current flow in response to different laser pulse durations was observed (Figure 6). The cells alone, compared to the cells exposed to AuNRs (Figure 6b,c), showed lower levels of, or no observable membrane current, when irradiated with varying pulse durations. The membrane depolarization of the neurons exposed to plasmonic nanoparticles always started at the onset of PNF activation. The action potential of the AuNRs-neuron was directly evoked by a single, 25-ms duration laser pulse, while subthreshold depolarizing potentials were seen at 1 and 10 ms laser pulses (Figure 6d).

Carvalho et al. [38] demonstrated the effect of PNFs on neuron action potential firing behavior. The aim of this study was to design a system that could deliver light’s energy locally to the neuron. Spherical gold nanoparticles (AuNP) with a diameter of 20 nm were used as the PNFs. AuNP targeting specificity was enhanced by conjugates Ts1, TRPV1ab and P2X3ab antibodies attached to the AuNPs; these antibodies or markers work as follow: Ts1 can bind to voltage-gated sodium channels, TRPV1ab antibodies targeting the TRPV1 ion channel, while P2X3ab targeting the P2X3 receptor ion channel. The results show that the surface modification enhanced neuron-PNF interaction and decreased the required concentrations of AuNP to create an action potential. Figure 7 illustrates the experimental setup, in which Dorsal root ganglion (DRG) neurons were used as the cell model, being patch-clamped in the whole-cell configuration. A 532-nm laser pulse was used to activate the PNF, and a theta capillary was used to add the AuNP via perfusion through one side, while the other side was used to washed away the AuNP by perfusion of fresh buffer.

In conclusion, this work showed that the PNFs triggered action potential events when their corresponding surface plasmon properties were activated. The specificity to generate action potentials was enhanced through the local delivery of surface-modified PNF to the cell membrane, although non-functionalized AuNPs show the ability to trigger an action potential, the AuNPs washed away from the cytoplasm quickly (Figure 7C). In addition, the results indicated that the PNF concentration required to produce the action potential with surface modified AuNRs is less than that required with non-modified surfaces, and washing and media exchange did not affect the ability of surface modified AuNRs to induce the action potential (Figure 8).

The second approach, used to evaluate PNF as a neural stimulator, was established by a 2D direct contact model and reported by our group recently [9,39]. In this model, we evaluated the interactions between the cell and the PNF at the cellular membrane’s outer surface. This method evaluates the interaction between the cells and the substrate, which starts at the ECM level. We [9] designed a two-dimensional (2D) system based on incorporation of PNFs (AuNRs) as an active layer on a plastic substrate (Thermanox coverslips) to test the stimulation of human mesenchymal stem cells (hMSCs) and investigate the neurogenic potential of these cells. The PNF coating was achieved through strong adhesion between the positive terminal of the functionalized AuNRs and the negative surface charge of the modified plastic substrate, as shown in Figure 9.

We found that the PNF system used was biocompatible as determined by assessment of cell morphology, immunohistochemistry staining, and expression of S100β, and glial fibrillary acidic protein (GFAP). Schwann cells maintained their integrity when seeded on the PNF system, and WST-1 assay showed a 14% increase in cell viability on the PNF system, as compared to the control. The goal of this experiment was to determine if neurogenic enhancement of the hMSCs occurs on this PNF system. Although this work did not show any activation of the PNF’s surface plasmon, it clearly showed that the PNFs in the form of AuNRs accelerated the differentiation of the hMSCs into a neural lineage. Human MSCs develop neural-like characteristics within 24 h of seeding. The expression patterns of Vimentin, GFAP, and S100 β were detected as early as 24 h after differentiation, while the same expression pattern was observed in the control, i.e., when the cells were seeded on tissue culture polystyrene substrate after 6 days (Figure 10).

Eom et al. [15] demonstrated a PNF stimulation strategy based on injection of AuNRs in a rat sciatic nerve model. Briefly, the experiment (Figure 11) consisted of local delivery of a 1-μL PNF AuNRs (with longitudinal plasmon resonance peak at 977 nm) using a microprocessor-controlled injection system. The treated area then was exposed to a light source (fiber-coupled laser diode) positioned at 50 µm. An electrical extracellular recording system was used for real-time monitoring of compound nerve action potentials in the presence and absence of PNF. Enhanced neural responses were recorded for nerves treated with PNF, while the control (no PNF) were 6 times less responsive when both preparations were exposed to an identical laser stimulation (0.641 J/cm^2^) (Figure 11b). This difference in neural responses likely resulted from the ability to deliver laser energy to the target tissue in a precise and efficient way through the excitation of the PNFs’ localized surface plasmons, which, in turn, produced local heat, leading to depolarization of neuron cell bodies and axons inside the nerve bundles.

## 3. Plasmonic Nanofactor as Inhibitor for Neural Activities

Directly and reversibly preventing neural activity without causing genetic modification is key for investigating the complex neural interactions behind how the brain behaves, as well as those behind the brain’s pathological conditions [17]. As mentioned before, early PNF exposure can act as an inibitor of neural activity; this function depends on the heat produced by irradiated light. Furthermore, the degree of inhibition can be tuned based on the level of PNF activation [40,41]. Understanding how the active plasmonic surface produces heat at nanoscale resolution will determine the level of excitability of neural circuits to be tuned in the nanoscale domain. This section highlights studies that explored the use of PNFs as inhibitors of neural activity. The basic procedure for achieving inhibition of neural activity might be attribtued to the interference between the PNF and the electrical preactivated neural network. The procedure can be summarized as following: First the cells were electrically stimulated to produce an action potential, followed by activation of PNF, as shown in Figure 12.

Kang et al. [18] successfully developed methods for fabricating a 2D PNF structure by optimizing an inkjet printable device with micro-pattern resolution to form a non-continuous coating. The design used in this work was based on adherance of the functionalized PNF (AuNRs) on a functionalized substrate. Briefly, the group modified the non-printable substrate with alternating layers of positive and negative polyelectrolytes through layer-by-layer (LbL) dip coating. They used poly(allylamin hydrochloride) as the positive electrolyte solution and poly(4- styrenesulfonic acid) as the negative. The coating of functionalized AuNRs was stabilized over the substrate through an electrostatic force between the surface charge of the AuNRs and the modified substrate; the best coating profile was achieved by negatively charged AuNRs printed on a positive electrolyte substrate. The printed PNF was activated by illumination the surface with an NIR laser in a continous mode, then the plasmonic activity was recorded as a function of heat generated by the PNF during and after the illumnation period, using an infrared camera (Figure 13). A clear change in plasmonic activity was recorded as a function of the illumination period.

To validate the biofunctionality of the printed PNF surface, microectrode array (MEA) chips were used as a base for printing PNF, then hippocampal neurons were cultured for 15 days over the printed surface to form a neural network. The goal of this work was to investigate whether PNF activation can alter synchronized neural network activity. A non-continuous coating of PNF printed in the center of the substrate activated only a small area of the entire channel recording area, as shown in Figure 14. When the printed PNF was activated with (138 mW/mm^2^), the synchronized neural network activity was suppressed over the PNF domain, with maximun suppression occurring at the center of the domain, as shown in Figure 15b,c. However, many of the active channels in the non-printed domain showed similar reduction inspike rate when the IR laser was turned on.

Another example of a neural network-PNF surface approach used for investigating the in vitro effects of PNF on neural behavior was reported by Lee et al. [19]. Figure 15 illustrates their general procedure. In this study, the researchers fabricated a plasmonic nanofilm via thermal evaporation of gold, whose plasmonic effects are detected in the NIR wavelength. Different film thicknesses (2, 5, 7.5, 10, and 20 nm) were produced in order to optimize the maximum plasmonic effect. To examine the plasmonic activity of the coating, an IR camera was used to track thermal activity of the samples under an NIR laser with a 785-nm continuous wave with 5-mm diameter circular illumination area; the power density was varied from 3 to 21 mW/mm^2^. The photothermal behavior of the PNF was dependent on the thickness of the coating and the laser profile (Figure 16).

Based on the thermal data, the group selected two coating thickness (2- and 5-nm) of PNF films deposited on MEA chips as the best plasmonically active surfaces for neural moderation. Hippocampal neurons were used as a network model. The results proved the biocompatibility of the PNF coating; the cells were healthy and successfully formed neural networks after 12 days of in vitro culturing. The researchers determined the changes in neural activity over the 2- and 5-nm-thick PNF films as a function of NIR laser intensity. They also reported that activating the PNF films led to the inhibition of the neural network activities, with maximum inhibition occurring when the 5-nm-thick PNF film was exposed to a 21-mW/mm^2^ NIR laser. Moreover, the inhibition phenomenon of the cultured neurons was reproducible without any noticeable damage to the integrity of the neurons.

Cell-PNF surface interaction demonstrates an inhibition of neural network activity when the PNF is activated, making it useful to investigate the mechanisms by which active PNF produces neural inhibition. Yoo at al. [17] did so by attaching PNFs to the cell membrane of hippocampal neurons (isolated from prenatal rat brains) through incubation of the cells with 10 μg/mL of PNF (Figure 17A). Hippocampal neuronal networks were cultured on a multichannel MEA chip, then the PNF were added with different incubation times (1, 3, 6, and 9 h). AuNRs were used as the PNF model (AuNRs functionlized with SH-PEG-NH_2_ with longitudinal plasmon resonance at 785 nm). The team investigated whether the activation of PNF influenced spontaneous neuronal activity of the hippocampal neurons. First, the spontaneous activity of the neurons was investigated under NIR irradiation or GNR treatment alone as a control. No effect on neural activity was detected in either case (Figure 17B). Upon activation of PNF with an IR laser pulse, the recorded spike rate significantly decreased as a function of increasing incubation time. The maximum inhibition of neural activity was achieved with 9 h of incubation, as shown in Figure 17C,D. Moreover, the degree of inhibition could be tuned by changing the laser power characteristics. For example, increasing the laser power from 0 to 15 mW/mm^2^ increased the inhibition proportionally from 0 to 89.6% (Figure 17E,F). This result shows that increasing the degree of PNF activation produces more heat, leading to a pronounced suppression of neural activity.

The team also studied the use of active PNF as a long-term neural modulator. After 9 h of incubation with PNF, hippocampal neuronal networks were exposed to NIR irradiation for varying lengths of time. The results indicated the ability of PNF to act as a long-term inhibitor of neural network activities (Figure 17G). Importantly, the long-term PNF stimulation had no apparent negative effect on spike rates and shapes (Figure 17H).

## 4. Conclusions and Future Insights

Controlling the cell environment at the nano- and micro-levels allows scientists to develop new methods and tools to enhance the field of regenerative medicine. The plasmonic ECM is one promising method to control neuronal activity and function using nanoscale engineering to customized surfaces. The two most important features of plasmonic ECMs are the ability to modify and tune the surface chemistry [9], and the ability to remotely activate surface plasmon for deep tissue treatment [15]. It has been demonstrated that plasmonic ECM can be tuned to enhance cell-based therapies, which in turn might stimulate or inhibit neural activity. The effects of these plasmonic nanostructures on neural activity could be expected to vary significantly. Stimulation or inhibition of the neural activity could be dependent on specific aspects of the biological systems, along with the laser excitation power and energy, or optical characteristics of the nanostructures. Based on these variable aspects becomes evident the possibility to tune the biological effects by observing stimulation or inhibition of the neural activity, based on various situations. First, activated –PNF under laser excitation could result in the promotion of neural activity when the neuronal networks don’t express spontaneous activity [42], or, second, activated –PMF with optimized laser energy might inhibit the neural activity in cases of neuronal networks expressing spontaneous activity, as reported previously [40,41]. The ability of plasmonic ECM to stimulate and enhance cell behavior can lead to new therapeutic devices that can treat central nervous system injuries by regulating the propagation and differentiation of neuronal stem cells and by modulating neural network activity. Furthermore, the use of a plasmonic ECM to suppress neural network activity could lead to a new therapeutic approach and to greater understanding of complex neural diseases, such as Parkinson’s and Alzheimer’s. Finally, this research may lead to innovative pain managment devices to alleviate chronic neuropathic pain.

The heat produced by activated surface plasmon is the most likely cause of the changes in neural network activity when cells are in contact with a plasmonic ECM. However, a clear scientific understanding of this phenomenon has not yet emerged in the literature. More investigations are warranted, especially into the cell signaling mechanisms. If the heat production hypothesis is indeed true, how would changing the plasmonic surface characteristics affect cell behavior or time-dependent changes in temperature? Additionally, how is neuronal activity generally affected by the electrical field produced by the surface plasmon? These questions remain to be answered in the future in this exciting research avenue.

## Figures and Tables

**Figure 1 nanomaterials-09-01029-f001:**
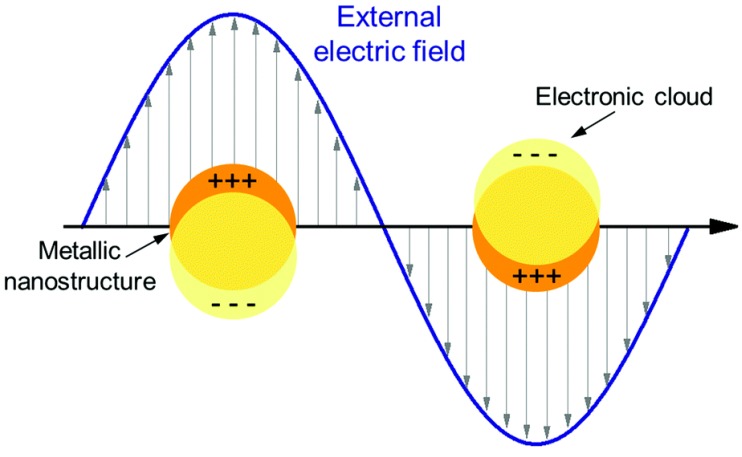
Plasmonic behavior of a metal surface, showing localized surface plasmon formation in the presence of an electromagnetic wave. Reproduced from [25]. Published by The Royal Society of Chemistry, 2017.

**Figure 2 nanomaterials-09-01029-f002:**
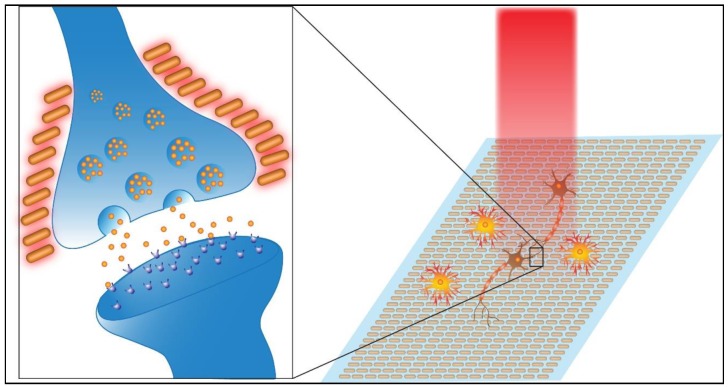
Events that could occur during the activation of plasmonic nanofactors (PNF). The insert shows that Ca^2+^ channels could be opened as a result of local heat generated by PNF, which leads to an increase in the Ca^2+^ concentration inside the neural terminal and the subsequent release of excitatory neurotransmitters that can depolarize the postsynaptic membrane, increasing the likelihood of action potential generation.

**Figure 3 nanomaterials-09-01029-f003:**
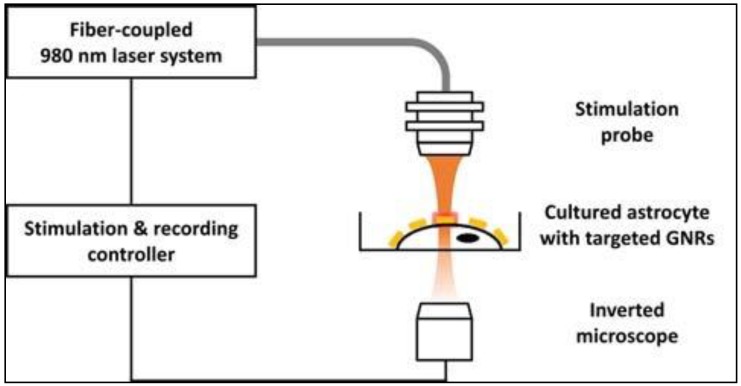
Experimental setup for cellular stimulation and calcium-sensitive dye imaging. Reproduced with permission from [33]. John Wiley and Sons, 2017.

**Figure 4 nanomaterials-09-01029-f004:**
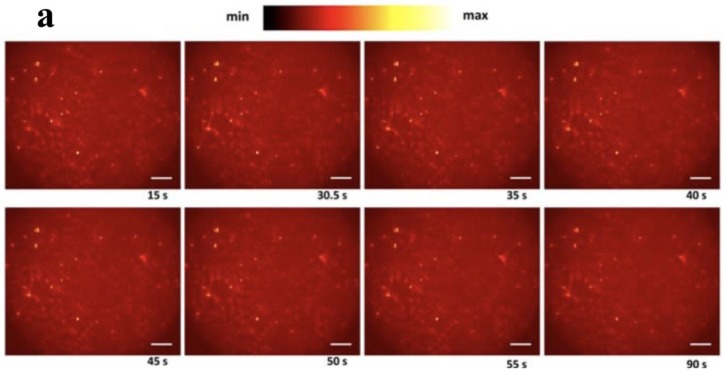
Characterization of near infrared (NIR) stimulation in the absence and presence of PNF. (**a**) Ca^2+^ images at 15 s, 30.5 s, 35 s, 40 s, 45 s, 50 s, 55 s, and 90s without AuNR. (**b**) Ca^2+^ images at 15 s, 30.5 s, 35 s, 40 s, 45 s, 50 s, 55 s, and 90 s with AuNR. (**c**) Intracellular Ca^2+^ waves profile presented before and after optical pulse (yellow line) in the presence of a nonbinding AuNR system. (**d**) Intracellular Ca^2+^ waves profile before and after optical pulse (yellow line) with a binding AuNR system. Reproduced with permission from [33]. John Wiley and Sons, 2017.

**Figure 5 nanomaterials-09-01029-f005:**
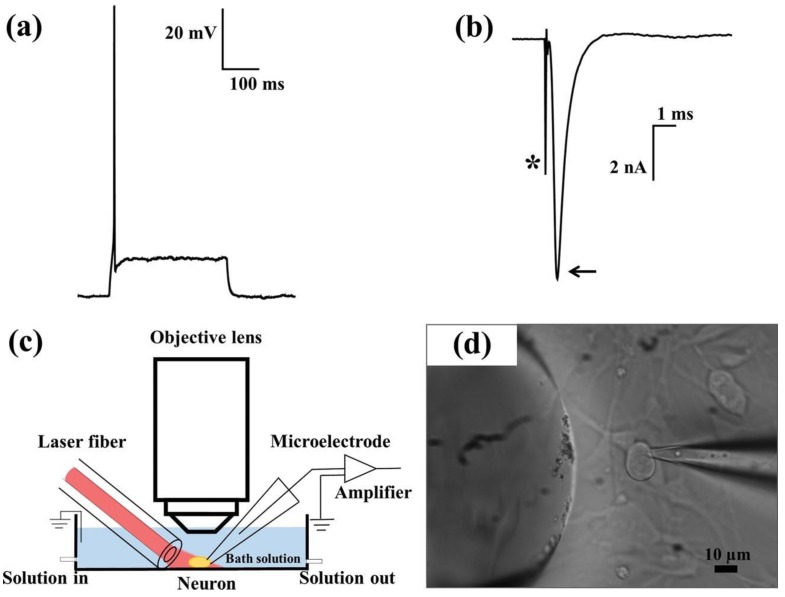
The typical response of a healthy neuron, held by whole-cell patch clamping, when exposed to certain stimuli. (**a**) A single action potential caused by a depolarizing current injection; (**b**) fast sodium currents during membrane depolarization in voltage-clamp (arrow). The stimulating electrode artifact is indicated with an asterisk. (**c**) The experimental setup for simultaneous laser stimulation and whole-cell patch clamp recordings on a neuron. (**d**) Phase-contrast image showing a patched neuron. Right: microelectrode; left: optical fiber. Reproduced with permission from [23]. John Wiley and Sons, 2014.

**Figure 6 nanomaterials-09-01029-f006:**
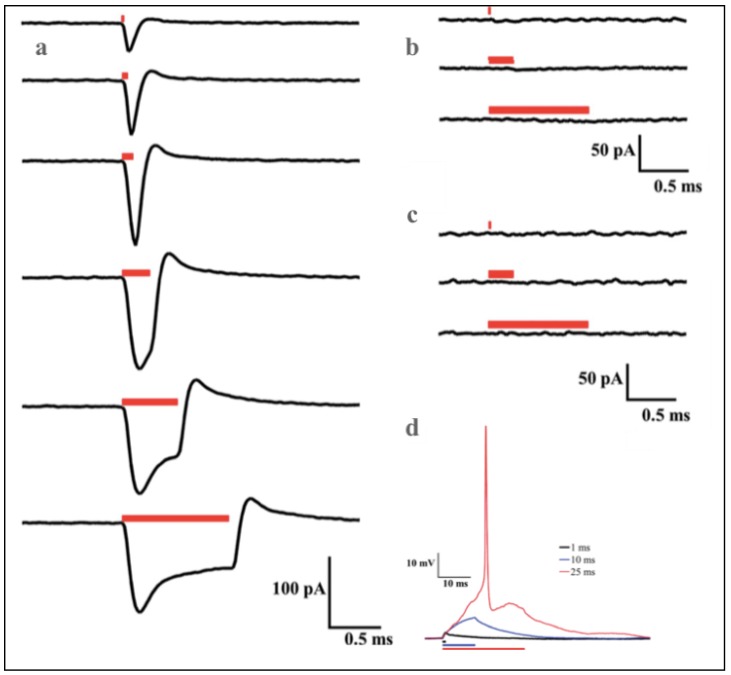
Representative averaged voltage-clamp data from (**a**) an NR-AN responding to laser pulses delivered over different durations (from top to bottom: 0.025, 0.05, 0.1, 0.25, 0.5, and 1 ms); (**b**) NS-AN receiving laser pulses for, from top to bottom, 0.025, 0.25, and 1 ms; and (**c**) a control neuron under similar conditions to (**b**). Bars indicate the timing and duration of laser pulses. All neurons were held at −73 mV. (**d**) action potential of AuNRs-neuron indirectly caused by one 25-ms laser pulse while subthreshold neuron potentials were being record at 1 and 10 ms laser pulses. Reproduced with permission from [23]. John Wiley and Sons, 2014.

**Figure 7 nanomaterials-09-01029-f007:**
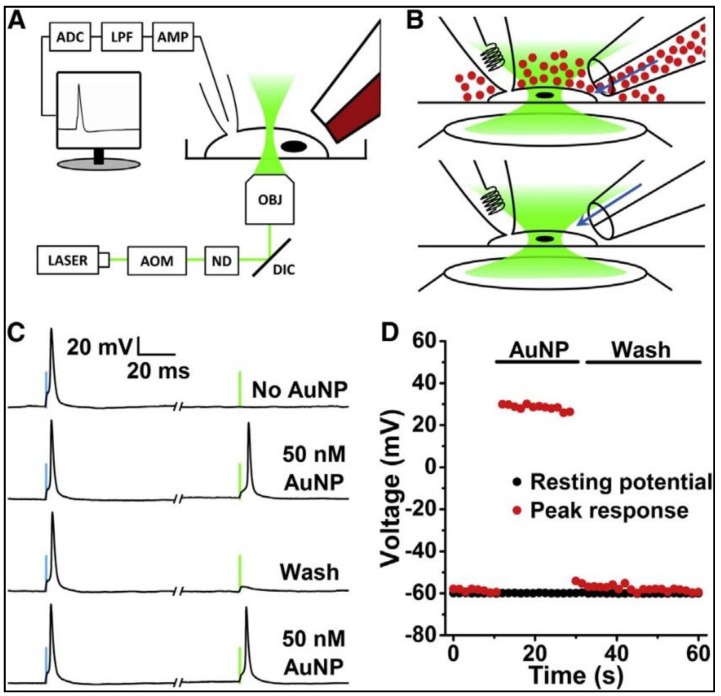
Action potentials were stimulated by a 532-nm laser when the cells were incubated with AuNPs. (**A**) Diagram of the experimental setup (not to scale). The cell was patch-clamped in a whole-cell configuration (pipette on the left) and adding AuNP and perfusion-washing the cells were performed via the theta capillary (right). (**B**) AuNPs were perfused over a patch-clamped DRG neuron through one side of a theta capillary. After a satisfactory optical response was observed, fresh buffer was poured over the cell from the other side of the capillary to wash off the AuNPs. (**C**) Representative traces of current-clamped DRG cells firing action potentials, caused by 2 stimuli: A 300-pA, 1-ms current injection (left, blue bars) and a 174-mW, 1-ms, 532-nm laser pulse (right, green bars). At first, cells only responded to the electrical stimulus, but the addition of AuNPs made the cells sensitive to light. Washing removed enough AuNPs from the cell to prevent the laser from triggering an action potential. The cells regained their optical sensitivity when a second dose of AuNPs was added to the bath. (**D**) The washing showed that the laser effect relies on having a sufficient AuNP concentration near the cell. After the cells were washed, the laser effect vanished in seconds. *Abbreviations:* AOM: acousto-optic modulator; ND: Neutral density filters; DIC: dichroic mirror; OBJ: Microscope objective; AMP: Amplifier; LPF: Low-pass filter; ADC: Analog-to-digital converter. Reproduced with permission from [38]. Elsevier, 2015.

**Figure 8 nanomaterials-09-01029-f008:**
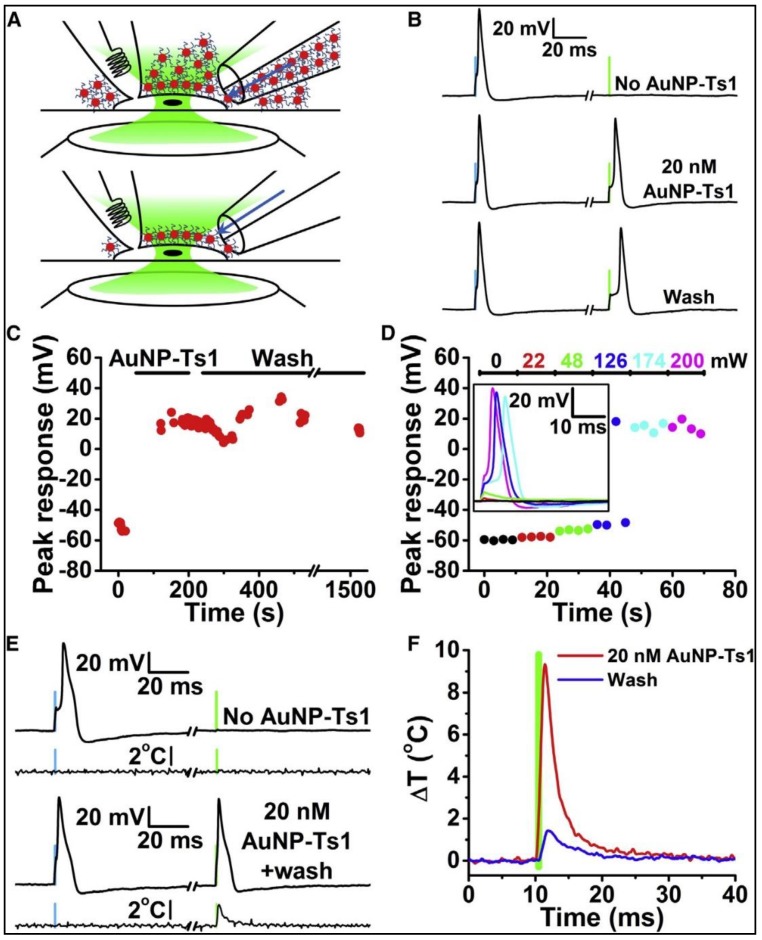
Optical stimulation of DRG neurons with AuNP-Ts1. (**A**) The same procedure used with non-functionalized AuNPs could not wash AuNP-Ts1 off of the cell. (**B**) At first, DRG neurons responded only to electrical stimuli (blue bars: 500 pA, 1 ms), but adding AuNP-Ts1 made the neurons sensitive to optical stimuli (green bars: 174 mW, 532 nm, 1 ms). (**C**) Even after 20 min of continual washing, AuNP-Ts1-labeled neurons were still optically sensitive. (**D**) Increasing the laser power increased cell depolarization. As a result, action potentials were triggered when the cell’s threshold voltage was reached. 174 mW was the required voltage to reliably stimulate this particular neuron. Inset: sample traces for each laser power. (**E**) Without AuNP-Ts1 (top 2 traces), no temperature changes were seen 2 mm away from a DRG cell during both electrical (blue bar) and optical (green bar) stimulation. However, when AuNP-Ts1 were added and excess nanoparticles were washed away (bottom 2 traces), optical stimulation was able to measurably increase the temperature. Temperature traces are single recordings with noise subtraction and are filtered at 1 kHz. (**F**) When nanoparticles were in the bulk solution (before washing), a significant temperature change was seen. After active perfusion washing, only the nanoparticles tightly bound to the surface were left. The cell was still optically excitable, but the temperature change 2 mm from the cell decreased dramatically (Green bar shows the optical stimulus). Reproduced with permission from [38]. Elsevier, 2015.

**Figure 9 nanomaterials-09-01029-f009:**
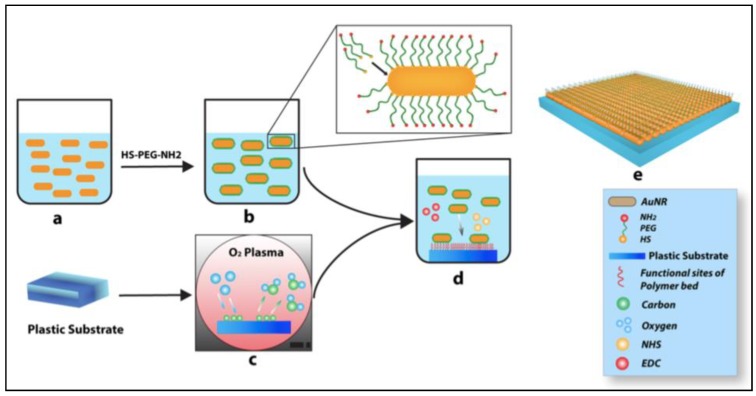
Protocol used to fabricate PNF two-dimensional (2D) system. (**a**) AuNRs preparation (**b**) Functionalization of AuNRs, (**c**) O_2_ plasma treatment to the plastic substrate, (**d**) Coating the plastic substrate with functionalized AuNRs (**e**) Washing with Di water, and then treatment with ethanol and UV for sterilization. Reproduced with permission from [9]. Springer Nature, 2014.

**Figure 10 nanomaterials-09-01029-f010:**
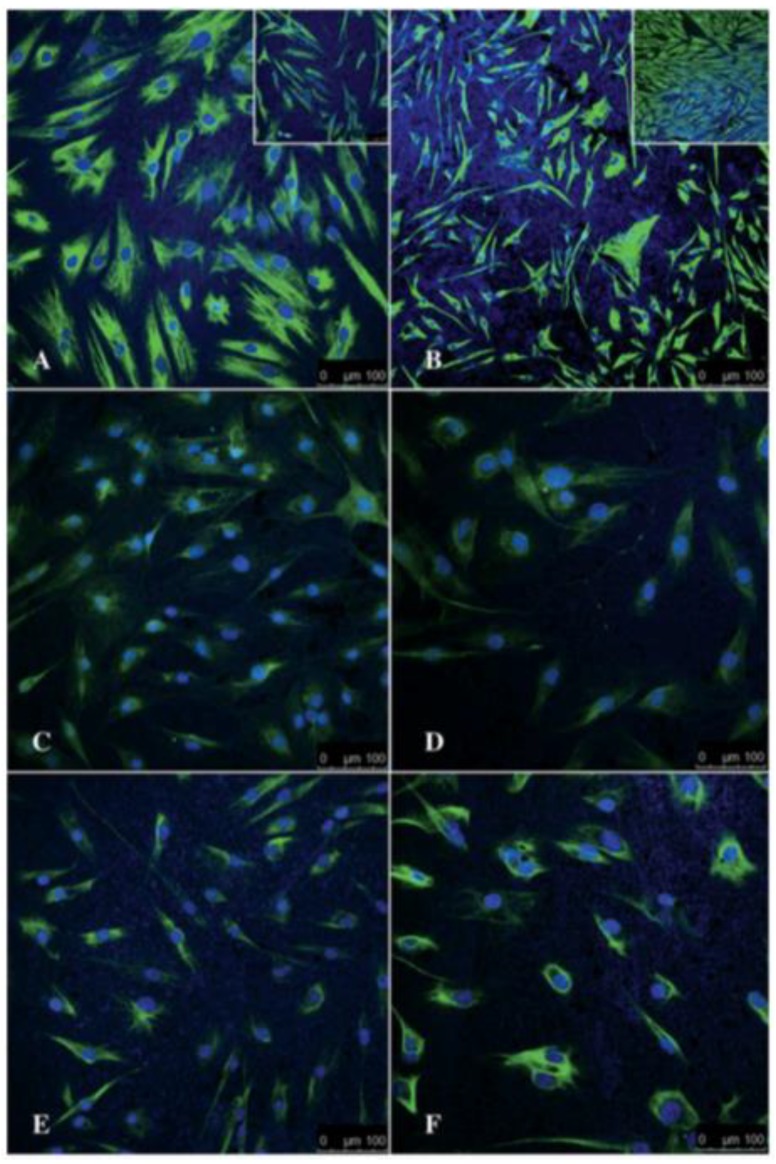
Representative 2D confocal microscopy images of hMSCs differentiating into cells of neural lineage on an AuNR substrate. Images show the expression of Vimentin, S100β, and GFAP at 24 h (**A**, **C**, and **E**, respectively) and 6 days (**B**, **D**, and **F**, respectively) after differentiation. The insets in A and B show the corresponding undifferentiated controls at 24 h and 6 days. Scale bar = 100 μm. Reproduced with permission from [9]. Springer Nature, 2014.

**Figure 11 nanomaterials-09-01029-f011:**
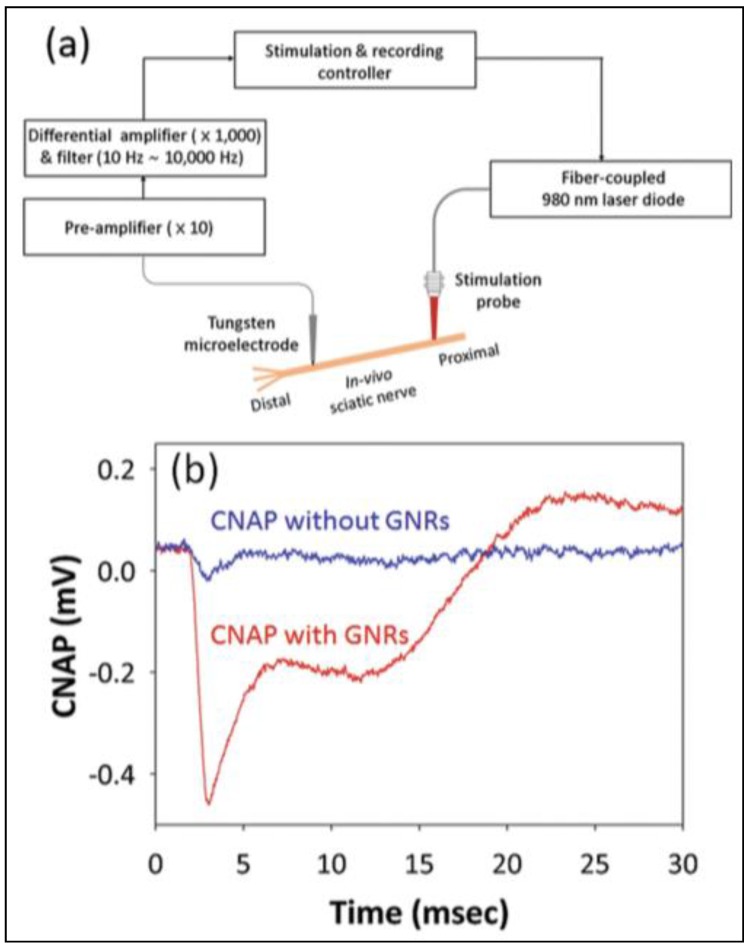
(**a**) a schematic of the experimental setup of in vivo recording of the rat sciatic nerve reactions to PNF activation. (**b**) CNAP curves recorded after laser exposure with and without PNF. Reproduced with permission from [15]. John Wiley and Sons, 2014.

**Figure 12 nanomaterials-09-01029-f012:**
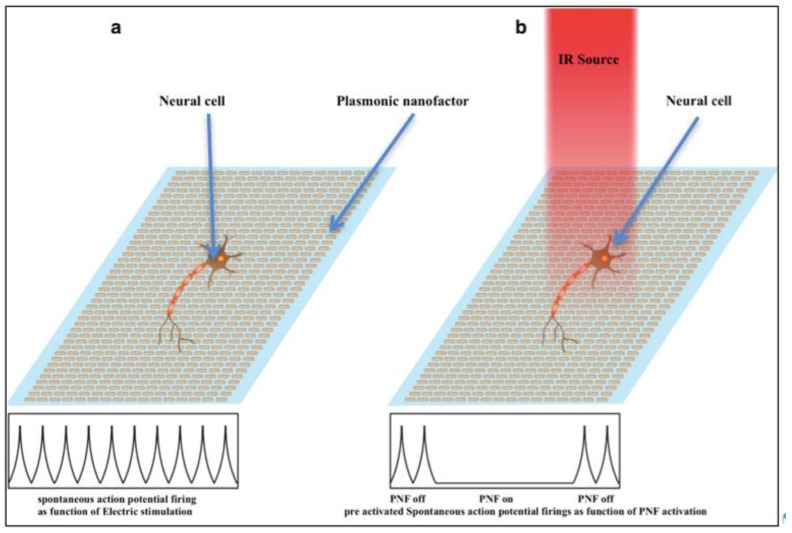
The general procedure for neural activity inhibition by PNF. (**a**) Preactivated neural cell by electric stimulation; (**b**) neural cell behavior under the influence of turning on and off PNF.

**Figure 13 nanomaterials-09-01029-f013:**
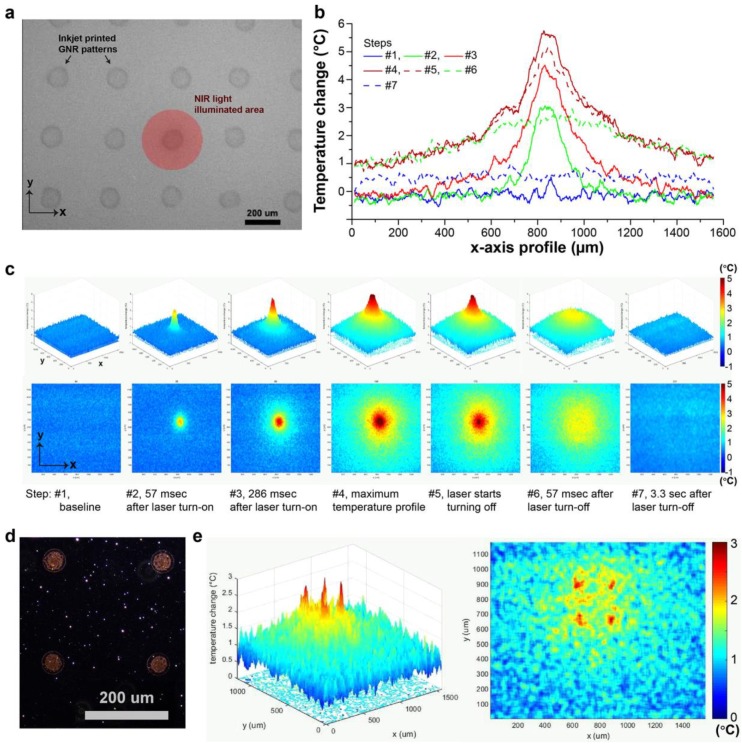
Heat generation by inkjet-printed micro-thermo-plasmonic (μTP) heaters. (**a**) Experimental setup: inkjet-printed GNR μTP pattern array on PAH+-coated glass coverslip via a 50-μm inkjet nozzle with 350-μm drop spacing. An 808-nm NIR laser with a 1.38 W/mm^2^ power density is illuminated only on a single dot pattern. Image spatial resolution: 2.4 μm/px. (**b**,**c**) 2D and 3D profiles of temperature change around a μTP heater in (**a**). Gaussian profile heat source formed within 300 ms after the NIR laser was turned on (step 3); the overall temperature increased due to heat dissipation from the heater (step 3 to 4), and the heat source rapidly vanished when the laser was turned off (steps 5, 6). (**d**,**e**) Smaller μTP heaters (dark-field image) and their heat profiles at their highest temperature change (diameter: 45 μm, drop spacing: 250 μm, inkjet nozzle: 30 μm, droplet volume: ~5 pL, NIR laser power density: 1.16 W/mm^2^). Reproduced with permission from [18]. American Chemical Society, 2018.

**Figure 14 nanomaterials-09-01029-f014:**
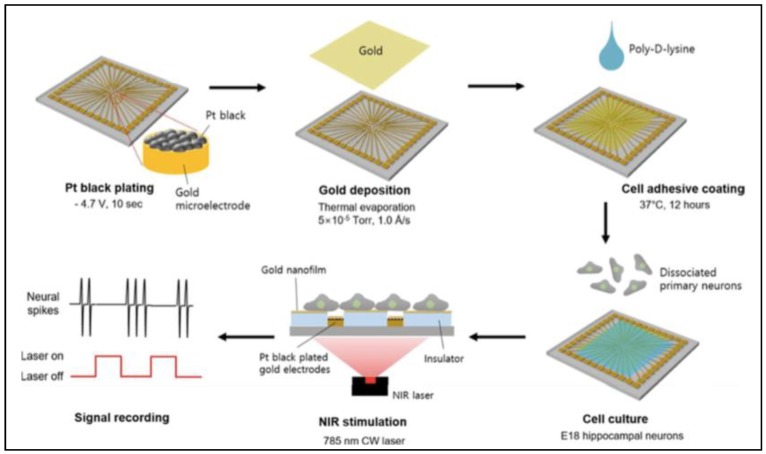
Illustration of the general procedures used to investigate the interactions between neural networks and PNF surfaces. Reproduced with permission from [19]. Royal Society of Chemistry, 2018.

**Figure 15 nanomaterials-09-01029-f015:**
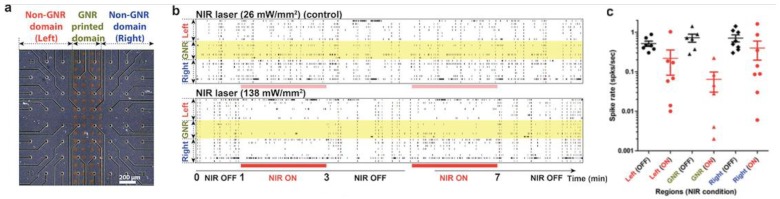
Synchronized neural network activity. (**a**) The cultured neuronal network (15 DIV) on an MEA chip, as imaged by dark-field microscopy. The chip has three sections—the GNR domain in the middle and the non-GNR domains on either side. Photothermally generated heat is only produced in the GNR domain. (**b**) The spike raster plots for all active electrodes (*n* = 22 for control, *n* = 21 for 138 mW/mm^2^) under NIR light stimulation with different light intensities (2 minutes of illumination followed by 2 min with the light off, repeated 5 times; the light illumination time window is indicated by color bars below each plot). (**c**) Average spike rate of active channels, shown according to domain and light illumination conditions (mean ± SEM). Reproduced with permission from [18]. Copyright American Chemical Society, 2018.

**Figure 16 nanomaterials-09-01029-f016:**
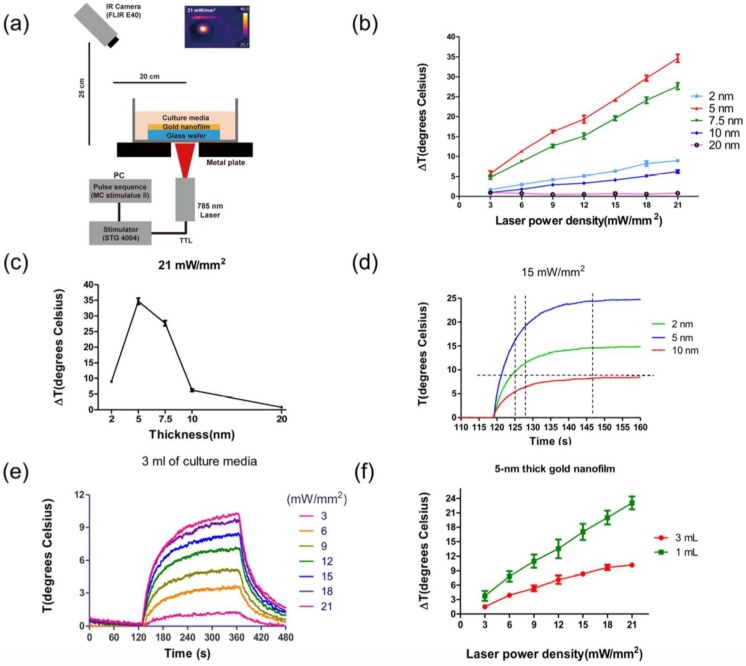
Photothermal behavior of PNF as a function of the thickness of the coating and the laser stimulation parameters. (**a**) Setup of experiment to measure the temperature change under NIR laser irradiation. (**b**) Temperature changes (ΔT) of gold nanofilm-coated glass wafer samples (*n* = 5) in air conditions when exposed to various laser power densities. (**c**) Maximum ΔT of various thicknesses of gold nanofilm samples (*n* = 5) at 21 mW/mm^2^. (**d**) Heat-up speed extraction of 2-nm, 5-nm, and 10-nm thick gold nanofilm-coated samples at 15 mW/mm^2^. (**e**) Temperature changes of 5-nm thick gold nanofilm sample in culture media. NIR light was turned on at around 120 s and was kept on for 120 s, then shut down immediately. (**f**) Maximum ΔT of the 5-nm thick samples (*n* = 3) in different volumes of culture media. Error bars represent standard deviation of measurement. Reproduced with permission from [19]. Royal Society of Chemistry, 2018.

**Figure 17 nanomaterials-09-01029-f017:**
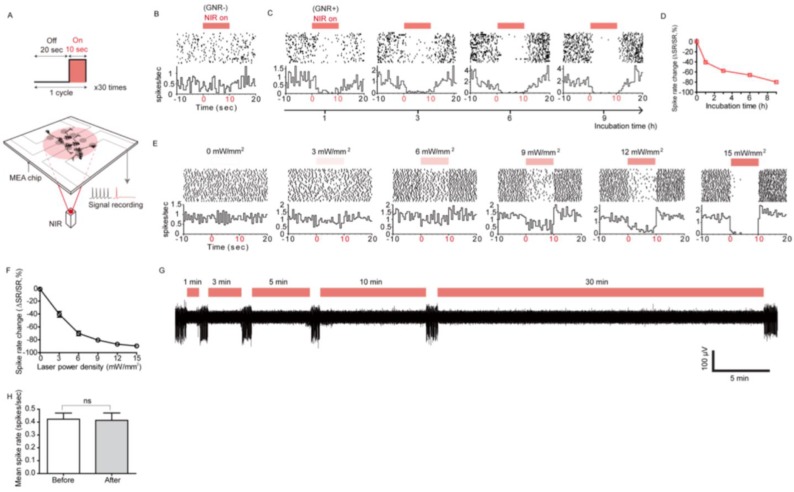
Photothermal modulation of spontaneous neural activity. (**A**) Schematic of repeated GNR-mediated photothermal stimulation of a neuronal network. Neurons were treated with 10 μg/mL of NH2-PEG-GNRs, thenirradiated with an NIR laser multiple times (785 nm, 0–15 mW/mm^2^). (**B**) Spike rates of untreated neurons when repeatedly irradiated with an NIR laser (15 mW/mm^2^). (**C**) NIR radiation (15 mW/mm^2^) pike rates of NH2-PEG-GNR-treated neurons that had varying incubation times. (**D**) A chart of GNR-treated neuron spike rates after the multiple NIR irradiations, shown in (**C**). One way ANOVA testing (*p* < 0.0001) showed that *n* = 264, 74, 97, 99, and 98 at 0 h, 1 h, 3 h, 6 h, and 9 h, respectively. The data point for 0h corresponds to no PNF treatment, as shown in (**B**). (**E**) Peri-event histograms and raster plots for GNR-treated neurons, incubated for 9 h, that were irradiated at multiple laser power densities. (**F**) Chart of spike rate changes from the experiments recorded in (**E**). One way ANOVA testing was conducted on the neurons tested in (**E**) for each laser power density. The results were as follows (*p* < 0.0001): *n* = 53 at 0 mW/mm^2^, 54 at 3 mW/mm^2^, 54 at 6 mW/mm^2^, 54 at 9 mW/mm^2^, 58 at 12 mW/mm^2^, and 58 at 15 mW/mm^2^. (**G**) A single trace of spike recording for different NIR irradiation periods (GNR incubation: 9 h, laser power density: 15 mW/mm^2^). (**H**) Mean spike rates of GNR-treated neurons before and after NIR irradiation (Data collected from the channels in which neuron activities were completely suppressed during NIR irradiation, two-tailed unpaired t test; *p* = 0.9064, *n* = 40 for each bar). Reproduced with permission from [17]. American Chemical Society, 2014.

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
