# Peer review of "Plasmonic Nanofactors as Switchable Devices to Promote or Inhibit Neuronal Activity and Function"

_nanomaterials, 2019, doi:10.3390/nano9071029_

Round 1
Reviewer 1 Report
The paper prepared by Alghazali et al, Plasmonic nanofactors as switchable devices to promote or inhibit neuronal activity and function, deserve to be published after minor improvment:
reference list must be update. In the current form present only 32 titles, only 10 from the last 2 years.
At least 40 papers were published according to Scopus on this subject only in the 2017 and 2018 and >10 in 2019.
After this minor revision, the paper can be published.
Author Response
Response To Reviewer 1
“reference list must be update. In the current form present only 32 titles, only 10 from the last 2 years. ”
Response: Thank you so much for this observation.. We agree with the reviewer and we have added additional more current references, shown below:
MOU, Z., YOU, M. & XUE, W. 2018. Gold nanorod-assisted near-infrared stimulation of bullfrog sciatic nerve. Lasers Med Sci, 33, 1907-1912.
Lee JW, Jung H, Cho HH, Lee JH, Nam Y. 2018. Gold nanostar-mediated neural activity control using plasmonic photothermal effects. Biomaterials 153:59-69
Yoo S, Park J-H, Nam Y. 2019. Single-Cell Photothermal Neuromodulation for Functional Mapping of Neural Networks. ACS Nano 13:544-51
de Boer WDAM, Hirtz JJ, Capretti A, Gregorkiewicz T, Izquierdo-Serra M, et al. 2018. Neuronal photoactivation through second-harmonic near-infrared absorption by gold nanoparticles. Light: Science & Applications 7:100
EOM, K., BYUN, K. M., JUN, S. B., KIM, S. J. & LEE, J. 2018. Theoretical Study on Gold-Nanorod-Enhanced Near-Infrared Neural Stimulation. Biophysical Journal, 115, 1481-1497.
SANCHEZ-RODRIGUEZ, S. P., SAUER, J. P., STANLEY, S. A., QIAN, X., GOTTESDIENER, A., FRIEDMAN, J. M. & DORDICK, J. S. 2016. Plasmonic activation of gold nanorods for remote stimulation of calcium signaling and protein expression in HEK 293T cells. Biotechnology and Bioengineering, 113, 2228-2240.
CARVALHO-DE-SOUZA, J. L., NAG, O. K., OH, E., HUSTON, A. L., VURGAFTMAN, I., PEPPERBERG, D. R., BEZANILLA, F. & DELEHANTY, J. B. 2019. Cholesterol Functionalization of Gold Nanoparticles Enhances Photoactivation of Neural Activity. ACS Chemical Neuroscience, 10, 1478-1487.
JOHANNSMEIER, S., HEEGER, P., TERAKAWA, M., KALIES, S., HEISTERKAMP, A., RIPKEN, T. & HEINEMANN, D. 2018. Gold nanoparticle-mediated laser stimulation induces a complex stress response in neuronal cells. Scientific reports, 8.
YADID, M., FEINER, R. & DVIR, T. 2019. Gold Nanoparticle-Integrated Scaffolds for Tissue Engineering and Regenerative Medicine. Nano Letters, 19, 2198-2206.
Reviewer 2 Report
In this review, effects of plasmonic metal nanofactors (PMF) on the neural (network) activities are discussed. Both promotion effects and inhibition effects are presented here, and it is difficult for me to understand what determines the promotion or the inhibition or what is promoted and what is inhibited. On one hand, Fig.2 shows the promotion of neural activity by the enhanced release of Ca2+ by local heat generation with IR irradiation to PMF. On the other hand, Fig.12 presents the suppression of neural activity. Here also IR irradiation was employed but for the PMF mounting pre-activated neural cells by electric stimulation. Apparently the “activity” that we discuss here is the electric current and not the proliferation rate or something else. If we do the same experiment without performing pre-activation, does the IR irradiation promote the neural activity?
Minor remarks
p.1 l.2 hasw → has
p.5 l.4-5 their the → either “their” or “the”
p.7 l.7 nonafactor-laser → nanofactor-laser
p.8 last l. mJ/mm-2 → mJ/mm2
p.19 l.3 PNF(AuNRs → PNF AuNRs
l.9 (0.641 J/cm2 → (0.641 J/cm2)
p.20 l.1 nonafactor → nanofactor
p.21 l.6 summarize → summarized
p.22 l.2 poly(allylamin hydro..)→ poly(allylamine hydro..)
l.3 styrernesulfonicacid) → styrenesulfonic acid)
l.8 illumnation → illumination
★Information of some references are incomplete; Refs. 11, 12, 20, 21, 27.
Author Response
Response To Reviewer 2
“Both promotion effects and inhibition effects are presented here, and it is difficult for me to understand what determines the promotion or the inhibition or what is promoted and what is inhibited. On one hand, Fig.2 shows the promotion of neural activity by the enhanced release of Ca2+ by local heat generation with IR irradiation to PMF. On the other hand, Fig.12 presents the suppression of neural activity. Here also IR irradiation was employed but for the PMF mounting pre-activated neural cells by electric stimulation. Apparently the “activity” that we discuss here is the electric current and not the proliferation rate or something else. “
Response:
Yes, for both Fig 2 and figure 12 we discuss the possible effect of PMF on the electric current of the neuronal networks, however we also describe and reported the possible effect of PMF on the differentiation of human mesenchymal stem cells into neural-like progenitors (1). The available literature suggests that the effects of these plasmonic nanostructures on neural activity can be quite variable. We have introduced text into the paper to further describe this particular point. We believe that the specific aspects of each biological system, along with the laser power, excitation energy or optical characteristics of the nanostructures could all play a role.
“If we do the same experiment without performing pre-activation, does the IR irradiation promote the neural activity?”
Response:
Since our manuscript is a review of the literature, we did not perform any of the studies described in our own laboratory.. However, the reviewer brings up a very significant issue and question. Based on what we have seen in the literature and our own experience in the field, two possible scenarios are as follows:
1- Activated –PMF with optimized Laser energy might lead to promotion of the neural activity in a situation where neuronal networks don't express spontaneous activity
2- Activated –PMF with optimized Laser energy might lead to inhibition of neural activity in a case where neuronal networks express spontaneous activity as reported previously (2; 3)
Based on our review of the available literature and ongoing studies, we anticipate being able to contribute to this area of science in the near future.
1. Alghazali KM, Newby SD, Nima ZA, Hamzah RN, Watanabe F, et al. 2017. Functionalized gold nanorod nanocomposite system to modulate differentiation of human mesenchymal stem cells into neural-like progenitors. Scientific reports 7:16654
2. Lee JW, Jung H, Cho HH, Lee JH, Nam Y. 2018. Gold nanostar-mediated neural activity control using plasmonic photothermal effects. Biomaterials 153:59-69
3. Yoo S, Park J-H, Nam Y. 2019. Single-Cell Photothermal Neuromodulation for Functional Mapping of Neural Networks. ACS Nano 13:544-51